# Leisure Screen Time and Food Consumption among Brazilian Adults

**DOI:** 10.3390/ijerph21091123

**Published:** 2024-08-26

**Authors:** Rayssa Cristina de Oliveira Martins, Thaís Cristina Marquezine Caldeira, Marcela Mello Soares, Laís Amaral Mais, Rafael Moreira Claro

**Affiliations:** 1Public Health Postgraduate Program, Medical School, Federal University of Minas Gerais, Belo Horizonte 30130-100, Brazil; nutrirayssaoliveira@gmail.com (R.C.d.O.M.); marmell.95@gmail.com (M.M.S.); 2Instituto Brasileiro de Defesa do Consumidor (Idec), Sao Paulo 01139-000, Brazil; lais.amaral@idec.org.br; 3Nutrition Department, Federal University of Minas Gerais, Belo Horizonte 30130-100, Brazil; rafael.claro@gmail.com

**Keywords:** television, screen time, food consumption, health surveillance

## Abstract

Background: Screen time, involving activities like watching television (TV), and using tablets, mobile phones, and computers (electronic devices), is associated with the consumption of unhealthy foods. This study aimed to analyze the association between prolonged leisure screen time and healthy and unhealthy food consumption indicators among Brazilian adults (≥18 years). Methods: Data from the National Health Survey (NHS), conducted in 2019 (n = 88,531), were used. Prolonged leisure screen time (screen time ≥ 3 h/day) was analyzed in three dimensions: watching TV; use of electronic devices; and total screen time (TV and electronic devices). Food consumption was analyzed in two dimensions: healthy (in natura and minimally processed foods) and unhealthy (ultra-processed foods). Poisson regression models were used to calculate prevalence ratios (crude and adjusted (PRa)) by sociodemographic factors (sex, age, schooling, income, area of residence, and race/color) and health factors (weight status, self-rated health, and presence of noncommunicable disease), to assess the association between prolonged screen time and food consumption indicators. Results: Among Brazilian adults, the prevalence of prolonged screen time was 21.8% for TV and 22.2% for other electronic devices for leisure. The highest frequency of watching TV for a prolonged time was observed among women, older adults, and those with a lower income and schooling. Prolonged use of electronic devices was more common among young adults and those with intermediate schooling and income. Prolonged screen time was associated with an unhealthy diet, due both to the higher consumption of unhealthy foods (PRa = 1.35 for TV, PRa = 1.21 for electronic devices, and PRa = 1.32 for both types) and the lower consumption of healthy foods (PRa = 0.88 for TV, PRa = 0.86 for electronic devices, and PRa = 0.86 for both). Conclusions: Prolonged screen time was negatively associated with the consumption of healthy foods and favored the consumption of unhealthy foods among Brazilian adults.

## 1. Introduction

Unhealthy food consumption, based on high consumption of ultra-processed foods, increases the risk of noncommunicable diseases (NCD), such as cancer and cardiometabolic multimorbidity [1]. The current dietary pattern worldwide, including in Brazil, is characterized by low consumption of in natura and minimally processed foods (such as grains, fruits, and vegetables), along with the excessive consumption of ultra-processed foods (such as snacks, sweetened drinks, and sausages) [2,3]. This shift is driven by the modern food system, leading to monotonous, low-nutrition diets that contribute to NCDs and global obesity [3]. NCDs account for 41 million deaths annually, or 74% of global deaths [4].

Sedentary behaviors include prolonged periods of sitting or lying down, with screen time being a particular concern in many studies [5,6,7,8]. These behaviors, including prolonged TV and computer or cell phone use, are generally linked to unhealthy eating habits [5,6,7]. Much of the concern in this regard is due to two mechanisms in food consumption: first, there is an increase in caloric intake through passive (often unconscious) consumption of food while sitting in front of a screen (in terms of quantity and quality of food, especially of ultra-processed foods) [5,6]; second, there is massive exposure to ultra-processed foods advertising [5,6,7,9]. It is worth noting that unhealthy eating habits combined with physical inactivity were responsible for 194 million healthy years of disability-adjusted life lost worldwide and 7.876 million deaths in 2021 [10].

The relationship between screen time and food consumption has been explored by several researchers worldwide [6,7,11,12,13,14]. The main sedentary behavior analyzed was prolonged TV watching [6,7,11,12,13], particularly due to the culture over the past decades of spending time in front of this type of screen while eating [7]. However, over the years, screen time on other electronic devices, such as computers, cell phones, tablets, and video games, has increased considerably, especially among younger individuals in the population [15,16,17]. In this context, although some studies are already analyzing the transition from prolonged TV viewing to other electronic devices, particularly among children and adolescents [8,14], our understanding of the relationship between different types of screen time and food consumption among adults is still incomplete.

Furthermore, previous studies have often examined the association between screen time and specific food consumption indicators, such as fruits, vegetables, snacks, or sweetened beverages, in isolation [6,7,11,12]. This narrow focus overlooks broader dietary patterns that emerge when multiple food groups are considered together. To address this gap, our study employs a simplified dietary scoring model, which has been utilized in various contexts to assess and monitor dietary patterns [18]. This approach not only enhances our understanding of how screen time influences overall diet quality but also fills a crucial gap in the literature by analyzing the association between screen time and comprehensive dietary patterns in adults.

Thus, this study aimed to investigate the relationship between prolonged exposure to screen time during leisure time (using different types of screens—TV, computer, cell phone, and tablet) and indicators of healthy and unhealthy food consumption.

## 2. Materials and Methods

### 2.1. Design and Population Sample

This was a cross-sectional study based on data collected from the National Health Survey (NHS) carried out in Brazil in 2019. The NHS is a nationwide household survey (with the Brazilian population, aged 15 years or older), conducted by the Ministry of Health in partnership with the Brazilian Institute of Geography and Statistics (Instituto Brasileiro de Geografia e Estatística—IBGE, in Portuguese) [19]. The NHS aims to collect information on the population’s health conditions, NCD morbidity surveillance, and their associated risk factors. Furthermore, it monitors the performance of the national health system in terms of access and use of available services and continuity of care [19].

The NHS relies on a clustered sampling strategy in three steps. The first step consisted of a census tract or a group of tracts, the primary sampling units (PSUs). The households were then selected within each census tract, the secondary sampling units (SSUs). Finally, within each household, a resident, aged 15 years or older was randomly chosen based on the list of residents made at the time of the interview. The estimated total sample size was 107,628 households, considering a non-response rate of 20%, a sample of 86,820 households or individual interviews was expected. A total of 90,846 individuals were interviewed. NHS 2019 data collection took place between August 2019 and March 2020 in Brazilian households, considering the sample distribution [19].

The sample collected is representative of the Brazilian population when the use of the sampling weight in the three sampling levels applied in the selection of individuals is considered [19]. Further information on the methodology is available in a specific publication [19].

The NHS 2019 was approved by the National Commission on Research Ethics (CONEP/MS, n° 3.529.376) [19] and the data are available for public access and use on the official IBGE website (https://www.ibge.gov.br/estatisticas/sociais/saude.html, accessed on 22 August 2024). The informed consent terms were signed by the interviewees. All informed consent terms are available on the NHS 2019 website [19].

The present study used a subsample of NHS 2019 data, consisting exclusively of adults (≥18 years) (n = 88,531).

### 2.2. Data Organization

The topics of central interest for the present study were those relating to lifestyle, with indicators of food consumption and prolonged screen exposure during leisure time. The sociodemographic characteristics and health conditions information complete the data of interest to this study.

Prolonged screen exposure during leisure time (exposure)

Prolonged screen exposure (≥3 h/day) during leisure time was investigated in three dimensions: time spent watching TV per day; time spent using a computer, a cell phone, and/or a tablet (electronic devices) per day; and exposure to TV and/or electronic devices. Prolonged exposure to TV considered individuals who reported spending three or more hours per day watching TV, through the question: “How many hours a day, on average, do you spend watching television? (<1 | 1 to <2 | 2 to <3 | 3 to <6 | ≥6 h | I don’t watch television)”. Prolonged exposure to electronic devices was identified in individuals who reported spending three or more hours per day using these devices during their leisure time, through the question: “In a day, how many hours of your free time (excluding work) do you usually use a computer, tablet, or cell phone for leisure? (<1 | 1 to <2 | 2 to <3 | 3 to <6 | ≥6 h | Does not usually use computer, tablet, or cell phone in free time)”. Prolonged screen exposure (regardless of the type) corresponds to the sum of time spent between two dimensions of screen time (TV and electronic devices).

Food consumption (outcome)

Food consumption was characterized by two indicators, one synthesizing the consumption of healthy food (based on the consumption of in natura or minimally processed foods) and the other, the consumption of unhealthy foods (based on the consumption of ultra-processed foods), based on a pre-established 24 h non-quantitative dietary recall. Both are validated indicators of the quality of food consumption [20,21]. The classification of the groups was based on the recommendations of the Dietary Guidelines for the Brazilian Population [22].

The following question was applied: “I’m going to ask you some questions about foods you ate yesterday. I’m going to start with natural or basic foods: Lettuce, cabbage, broccoli, watercress, or spinach; pumpkin, carrot, sweet potato, or okra; papaya, mango, yellow melon, or pequi; tomato, cucumber, zucchini, eggplant, chayote, or beet; orange, banana, apple, or pineapple; beans, peas, lentils, or chickpeas; peanuts, cashew nuts, or Brazil/Pará nuts”. Similarly, for ultra-processed foods consumption investigation, the following question was applied: “Now I’m going to list foods or industrialized products: soft drink; fruit juice in a box or can; powdered soft drink; chocolate drink; flavored yogurt; salty snacks, chips or crackers; cookies, stuffed cookie or packaged cake; chocolate, ice cream, gelatin, flan or other industrialized dessert; sausage, mortadella or ham; loaf, hot dog, or hamburger bread; mayonnaise, ketchup or mustard; margarine; instant noodles, powdered soup, frozen lasagna or other ready-to-eat frozen dish”.

The healthy food consumption score was calculated by adding the “yes” answers to the in natura food groups, which can vary from zero to seven on the day before the interview. Subsequently, the score was dichotomized, with the cut-off being the consumption of five or more groups of in natura or minimally processed foods [23] (respecting a previously established methodology [23]).

A similar approach was used to indicate unhealthy food consumption scores. The sum of “yes” answers for the consumption of ultra-processed foods can vary from zero to ten, on the day before the interview. This score was also organized in a dichotomized manner, based on the “yes” answers to the consumption of five or more groups of ultra-processed foods [20].

Control independent variables

To complement the analyses, the sociodemographic characteristics of the individuals were included: sex (male | female); age group (18–24 | 25–34 | 35–44 | 45–54 | 55–64 | 65 years of age or older); years of schooling (0–8 | 9–11 | ≥12 years); income (minimum wage—MW)(<1 | ≥1 to <3 | ≥3 to <5 | ≥5 MW); race/color (white | black/brown | yellow/Indigenous), and state of health (weight status, NCD, and self-rated health).

The weight status was identified by body mass index (BMI) calculated based on self-reported weight and height, according to the cut-off recommended by the World Health Organization (WHO) [24]. Absent values of weight and height were previously imputed using hot deck methodology [19]. The presence of NCDs was established based on the presence of one or more of the most common NCDs in Brazil, evaluated through the question: “Has any doctor ever diagnosed you with [name of the disease]? (Hypertension (high blood pressure) | diabetes| high cholesterol| heart disease| stroke| cancer| chronic kidney disease| chronic back problems)”. The negative self-rated health was identified through the answer “poor” or “very poor” to the question: “In general, how do you assess your health?”.

### 2.3. Data Analysis

The population was described by its distribution (%) and 95% confidence intervals (CI), according to the sociodemographic and health characteristics. A similar procedure was applied to describe the exposure (prolonged leisure screen time) and outcome (food consumption). Differences between the prevalence of prolonged leisure screen time and food consumption across sociodemographic characteristics were investigated through the overlapping of 95%IC (considering non-overlapping as a significant difference, a conservative approach).

Poisson regression models (with robust variance) were applied to analyze the association between the indicators of healthy and unhealthy food consumption (dichotomous dependent variable) and indicators of prolonged screen time during leisure time (dichotomous independent variables). Both crude (PRc) and adjusted prevalence ratios (PRa) were calculated, with adjustments made for sociodemographic (sex, age, schooling, income, race/color) and health (weight status, negative self-rated health, chronic disease) variables in the models.

Poisson models are a useful alternative to logistic regression for dichotomous outcomes in cross-sectional studies since the latter may overestimate the true association, especially when the prevalence of the dependent variable is high (over 10–20% [25], as observed in the results of the present study). The confounding variables (mentioned above) were selected based on their relationships with both the dependent and independent variables used in the regression models. In the adjusted models, all confounding variables were simultaneously included, allowing for the observation of the effect of the independent variable (prolonged screen time during leisure and its variations) on the dependent variables (related to food consumption). Three different models were employed for each outcome, independently investigating the target association for prolonged screen time in TV, in electronic devices and combining the use of both types of screens. This approach was chosen because it allows for the investigation of the influence of each screen type, enabling the identification of any differences between them.

All estimates considered NHS weighting factors (and complex two-stage sample) to represent the study population. The data were organized and analyzed using Stata Statistics software, version 16.1 (Stata Statistical Software Release 16.1. Stata Corporation, 2019). The significance was determined by 95% CI and *p* ≤ 0.05.

## 3. Results

The study population was composed mostly of women (53.2%) and black-brown individuals (55.3%). Just over a third of the study population was aged 35 to 54 years (38.0%), and almost half had up to 8 years of schooling (49.2%) and a per capita household income of less than 1 MW (47.5%). More than half of the study population was overweight (36.6% with pre-obesity and 22.0% with obesity), and more than a third had at least one NCD (37.5%). Only 5.8% rated their health negatively (Table 1).

About one in three individuals (37.4%) spent three hours or more per day in leisure screen activities. This was most frequent among women. Prolonged watching TV was more frequent among the older age groups (65 years or older, 31.6%) and those with 0 to 8 years of schooling (23.9%). A distinct scenario was observed in the case of the prolonged use of electronic devices, with the highest frequencies found in the younger age groups (18 to 24 years, 52.2%, gradually decreasing with the increase in age) and those with 9 to 11 years of schooling (32.8%). In general, the indicator of prolonged TV watching showed less variation between the sociodemographic and health variables than that related to the electronic devices (Table 2).

Nearly one in four individuals (23.7%) reported consuming five or more healthy food groups on the day before the interview (14.3%). This frequency increased with age (from 14.9% for the 18 to 24 years group to 28.3% for the 45 to 54 years group and remained stable in the subsequent age groups). The opposite scenario was observed for the indicator of unhealthy food consumption (varying from 24.6% for the 18 to 24 years group to 7.0% for the 65 years of age or older group). Schooling and income were also directly associated with the frequency of healthy food consumption. However, in the case of unhealthy foods, the highest frequency was observed in the 9 to 11 years of schooling, while no clear relation was found with income (Table 3).

In general, individuals with prolonged screen time (≥3 h/day) showed a lower frequency of the healthy food consumption indicator and a higher frequency of the unhealthy food consumption indicator, regardless of the type of screen (both PRc and PRa). The comparison among the adjusted coefficients also reveals values that are quite similar across the different types of exposure. Individuals who watched TV or used electronic devices for three or more hours per day had a 14% lower frequency of healthy food consumption (PRa = 0.86, *p* < 0.001) even after adjustments for sociodemographic and health variables. The frequency of unhealthy food consumption was 35% higher for TV (PRa = 1.35, *p* < 0.001), 21% higher for electronic devices (PRa = 1.21, *p* < 0.001), and 32% higher when any type of screen was considered (PRa = 1.32, *p* < 0.001) (Table 4).

## 4. Discussion

Using data from the 2019 NHS, which included a nationally representative sample of more than 88,000 Brazilian adults, we examined the association between prolonged leisure screen time and diet quality indicators. Approximately 37.4% of the study population spent prolonged periods in front of screens, regardless of the type (watching TV or using electronic devices). TV remains the preferred screen for older individuals (65 years or more) and those with 0 to 8 years of schooling, whereas electronic devices are favored by young adults (especially those under 35 years) and those with 9 to 11 years of schooling and higher schooling (12 or more years). Those with prolonged screen time had poorer quality diets, with their healthy food consumption being 10% lower, and the unhealthy one being 32% higher (35% more for prolonged TV viewers, 21% more for electronic device users).

Prolonged screen time is a major sedentary behavior associated with negative health outcomes, including higher mortality from NCDs, cardiovascular diseases [26], excess weight [27], reduced cognitive development in children [28], and depression in adults [29]. In Brazil, prolonged TV screen time has remained stable, with about 25% of adults spending more than three hours daily from 2016 to 2021, particularly the elderly and individuals with lower educational levels [15]. Simultaneously, screen time for other electronic devices increased from 19.9% to 25.5% among adults, especially younger adults (18–34 years old) with 9 to 11 years of education [15]. This trend is also seen in high-income countries like the U.S., where between 2003 and 2016, 65% of adults watched more than two hours of TV daily, and by 2016, about 50% spent over an hour on other screens [16]. In Canada, the percentage of adults who spent four or more hours a day in front of screens on a non-workday was 37% in 2021, and among the 94.5% of Canadians who used the internet in 2022, 28% spent more than 4 h per day in front of screens [17].

The concern over the relationship between prolonged screen time and food consumption is heightened by significant shifts in dietary habits [2,3], particularly due to increased access to ultra-processed foods at the expense of healthier options [3]. Prolonged screen time is linked to mechanisms influencing food choices, such as exposure to the marketing of ultra-processed foods, which can shape consumption patterns [7,12]. In Brazil, 90.8% of food and beverage ads on free-to-air TV channels in 2018 were for ultra-processed foods, often using aggressive marketing tactics [9]. This exposure extends to the internet [30], social networks [31], and delivery apps [32]. Furthermore, prolonged screen time also results in distraction and inattentiveness, reducing awareness of food intake and portion sizes [7,22].

A cross-sectional study among Brazilian adults investigated the link between prolonged TV time and food consumption from 2006 to 2014. Using a similar indicator of exposure to that employed in the present study (TV ≥ 3 h/day), it examined 10 food consumption indicators related to NCDs. Watching TV ≥ 3 h/day was inversely associated with protective foods and directly associated with high-risk foods [11]. However, the study was limited to adults in state capitals, only considered TV, and lacked comprehensive dietary indicators. Our findings support these results and advocate for including other screens and overall dietary quality indicators [20,33].

The results of our study have important public health implications. Identifying specific risk groups, such as women, the elderly, and individuals with lower levels of education and income, who are more likely to use screens for a long time and, consequently, may develop inadequate eating habits, is essential for monitoring risk groups for NCDs and their risk factors in the country [34]. These potential risk groups have been observed in other studies, where sociodemographic factors like age and gender were found to moderate the association between food consumption and screen time [7]. These findings suggest the need for targeted interventions that consider the sociodemographic characteristics of these groups to maximize the effectiveness of health promotion strategies [7]. However, further analysis is needed to clarify this as it was not the central focus of this study. It should be noted that such interventions could be effective in preventing NCDs, such as obesity, diabetes, and cardiovascular diseases, given the relationship observed in previous studies [26,27,28,29].

The negative impact of prolonged screen time on diet quality highlights the importance of public policies that encourage reducing screen time as a strategy to improve dietary patterns and, by extension, the health of the population. While prolonged screen time is only one aspect of the issue related to unhealthy food consumption in the context of the current food system [3], it remains a significant risk factor for several diseases [26,27,28,29]. In our study, we observed a direct association between the consumption of unhealthy foods and all types of screen time, as well as an inverse association with the consumption of healthy foods. Therefore, it is crucial to consider interventions that effectively reduce screen time and address current dietary patterns, as these measures are essential for tackling the broader public health challenges associated with both screen time and unhealthy eating habits. In a systematic review conducted in 2011 [7], this relationship was observed among specific food groups, such as the inverse relationship between TV time and fruit and vegetable consumption, and a direct relationship with snacks and sweetened beverages consumption among children, adolescents, and adults [7]. However, screen time on electronic devices has not been well reported [7], particularly among adults, largely due to the recent increase in the use of this type of screen by the population [15,16,17], with the association between the consumption of snacks, fried and sweet foods, and high screen time on electronic devices in children and adolescents [14] being more widely reported.

In this sense, emphasizing interventions to reduce screen time and sedentary behavior is crucial to addressing the broader public health challenges associated with screen time and unhealthy eating habits. Interventions aimed at reducing screen time and sedentary behavior often focus on both decreasing overall screen time [8] and promoting leisure physical activity [7,8]. The Brazilian Physical Activity Guidelines recommend moving for at least 5 min every hour to enhance quality of life [35]. Improve diets, strategies such as taxing ultra-processed foods, regulating advertising on TV and other devices, and encouraging healthy eating through subsidies can be effective [3,36]. Additionally, reducing the link between prolonged screen time and food consumption is reinforced by various studies [6,7,11,12,13,14] and Brazil’s dietary guidelines, which suggest avoiding eating in front of screens and making mealtimes a shared family experience at the table [22].

## 5. Study Limitations

The present study has limitations that should be considered. First, although the most recent NHS was used, it was collected in 2019, which may partially affect the results (especially the prevalence of electronic devices time). Recent studies with Brazilian adults living in all state capitals suggest that our findings may underestimate current (2024) prolonged electronic device time and that of total prolonged screen time in the population [15]. Next, although common in large health surveys, self-reported information is prone to bias. Additionally, the questionnaire’s 24 h non-quantitative dietary recall limits food consumption understanding to predefined items. The quantity of food consumed in each group is also not collected. Nevertheless, the questionnaire is validated for the Brazilian population and reflects dietary patterns [33]. Moreover, not all screen types were included in the NHS (e.g., video games, which may be particularly relevant for young adults), which may increase reported screen time. There is no information regarding the stratification of screen time per electronic device and the content consumed. Nevertheless, no evidence suggests strong impact of these limitations on the present results, as validation studies of the questions employed in the NHS show good results [37], similar to the indicator of diet quality employed [20,21]. Finally, the gradient observed in the prevalence of prolonged screen time in our results is consistent with that observed in other studies using more accurate methods and smaller population sizes [38].

Despite these limitations, this study has strengths. It includes data from over 88,000 adults, providing a representative sample of the Brazilian population, where 96% had televisions and 84% had internet access in 2019 [39]. The stability in TV screen time and increased use of electronic devices underscore the relevance of this investigation [15]. Our findings contribute to the understanding of screen exposure and food consumption by examining leisure screen time across TV and electronic devices and food consumption patterns, aligning with Brazilian Dietary Guidelines [22,33].

## 6. Conclusions

Our study identified an association between prolonged leisure screen time (TV and electronic devices) and food consumption (healthy and unhealthy foods) among the Brazilian population. Longer leisure screen time was negatively associated with the consumption of healthy foods and positively associated with the consumption of unhealthy foods.

## Figures and Tables

**Table 1 ijerph-21-01123-t001:** Distribution ^a^ of the Brazilian adult population, according to sociodemographic characteristics and health conditions. NHS, 2019.

Variables	Total		
	%	95% CI	
Sex			
Male	46.8	46.2	47.4
Female	53.2	52.6	53.8
Race/color			
White	43.3	42.5	44.0
Black/brown	55.3	54.5	56.0
Yellow/Indigenous	1.5	1.3	1.6
Age (years)			
18–24	13.9	13.4	14.4
25–34	18.1	17.6	18.6
35–44	20.2	19.8	20.7
45–54	17.8	17.4	18.3
55–64	15.0	14.7	15.5
65 or older	14.9	14.5	15.4
Schooling (years)			
0–8	49.2	48.5	50.0
9–11	34.9	34.3	35.6
12 or more	15.8	15.2	16.5
Income (minimum wage) ^b^			
<1	47.5	46.7	48.3
≥1 a <3	41.0	40.3	41.7
≥3 a <5	6.4	6.0	6.7
≥5	5.2	4.8	5.6
Weight status			
Eutrophic ^c^	39.0	38.4	39.7
Pre-obesity ^d^	36.6	36.0	37.2
Obesity ^e^	22.0	21.4	22.8
Noncommunicable disease ^f^	37.5	37.0	38.1
Negative self-rated health	5.8	5.5	6.0

CI 95%: 95% confidence interval; NHS: National Health Survey. ^a^ Weighted percentage to adjust the sociodemographic distribution of the NHS sample to the distribution of the general population (see Methodology). ^b^ Minimum wage in 2019: R$998; excluded data from non-responders. ^c^ BMI < 25 Kg/m^2^ and BMI ≥ 25 Kg/m^2^. ^d^ <30 Kg/m^2^. ^e^ BMI ≥ 30 Kg/m^2^. ^f^ Presence of one or more chronic diseases (high blood pressure, diabetes, dyslipidemia, heart disease, stroke, cancer, kidney failure chronic disease). n = 88,531.

**Table 2 ijerph-21-01123-t002:** Prevalence ^a^ of adults reporting screen time ≥ 3 h/day for each type of screen in the entire population and according to strata defined by sociodemographic characteristics and health conditions. NHS, 2019.

Variables	Exposure to TV			Exposure to Electronic Devices			Total Screen Time ≥ 3 h/day		
	≥3 h/day			≥3 h/day					
	%	95% IC		%	95% IC		%	95% IC	
Sex									
Male	20.5	19.8	21.3	21.9	21.1	22.7	35.9	35.0	36.8
Female	22.9	22.2	23.6	22.5	21.7	23.3	38.7	37.8	39.6
Race/color									
White	20.8	20.0	21.6	21.6	20.7	22.5	36.2	35.2	37.2
Black/brown	22.6	22.0	23.3	22.6	21.9	23.3	38.3	37.5	39.1
Yellow/Indigenous	19.3	15.6	23.1	23.7	18.0	29.3	38.2	32.5	43.9
Age (years)									
18–24	19.2	17.6	20.8	52.2	50.1	54.3	59.2	57.2	61.3
25–34	21.1	19.8	22.4	36.0	34.5	37.4	46.1	44.6	47.6
35–44	17.6	16.5	18.7	21	19.8	22.2	32.2	31.0	33.5
45–54	19.5	18.3	20.7	12.3	11.4	13.2	27.7	26.3	29.1
55–64	23.6	22.5	24.7	8.8	8.0	9.6	28.4	27.2	29.6
65 or older	31.6	30.4	32.8	4.5	3.9	5.1	34.1	32.9	35.3
Schooling (years)									
0–8	23.9	23.1	24.6	13.3	12.6	13.9	32.5	31.7	33.4
09–11	21.5	20.6	22.4	32.8	31.7	33.9	44.9	43.8	46.0
12 or more	15.9	14.8	17.0	26.5	25.1	28	35.8	34.2	37.3
Income (minimum wage) ^b^									
<1	22.1	21.3	22.8	21.8	21.0	22.6	37.0	36.1	37.9
≥1 a <3	22.2	21.4	23.0	22.2	21.3	23.0	37.9	36.9	38.8
≥3 a <5	21.3	19.4	23.1	25.7	23.6	27.7	40.0	37.8	42.3
≥5	16.3	14.6	17.9	21.6	19.7	23.5	33.4	31.2	35.6
Weight status									
Eutrophic ^c^	19.8	19.1	20.5	23.0	22.1	23.9	36.7	35.7	37.6
Pre-obesity ^d^	21.5	20.6	22.3	20.2	19.4	21.1	35.4	34.4	36.4
Obesity ^e^	25.6	24.5	26.6	22.9	21.6	24.2	40.7	39.4	41.9
Noncommunicable disease ^f^	25.8	25.0	26.6	14.1	13.5	14.8	35	34.1	35.9
Negative self-rated health	26.9	25.0	28.7	11.2	9.8	12.7	34.1	32.0	36.1
Total	21.8	21.3	22.3	22.2	21.6	22.8	37.4	36.7	38.0

CI 95%: 95% confidence interval; NHS: National Health Survey; TV: television; electronic devices: computer, cell phone or tablet. ^a^ Weighted percentage to adjust the sociodemographic distribution of the NHS sample to the distribution of the general population (see Methodology). ^b^ Minimum wage in 2019: R$998; excluded data from non-responders. ^c^ BMI < 25 Kg/m^2^ and BMI ≥ 25 Kg/m^2^. ^d^ <30 Kg/m^2^. ^e^ BMI ≥ 30 Kg/m^2^. ^f^ Presence of one or more chronic diseases (high blood pressure, diabetes, dyslipidemia, heart disease, stroke, cancer, kidney failure chronic disease). n = 88,531.

**Table 3 ijerph-21-01123-t003:** Prevalence ^a^ of adults reporting healthy and unhealthy food consumption in the entire population and according to strata defined by sociodemographic characteristics and health conditions. NHS, 2019.

Variables	Consumption of Healthy Foods			Consumption of Unhealthy Foods		
	(≥5 Foods)			(≥5 Foods)		
	%	95% CI		%	95% CI	
Sex						
Male	22.9	22.2	23.7	15.7	15.0	16.4
Female	24.4	23.6	25.1	13.1	12.5	13.7
Race/color						
White	26.2	25.4	27.1	15.0	14.3	15.8
Black/brown	21.5	20.9	22.1	13.6	13.1	14.2
Yellow/Indigenous	31.2	26.2	36.7	18.5	14.0	24.1
Age (years)						
18–24	14.9	13.5	16.3	24.6	23.0	26.4
25–34	19.1	18.1	20.2	19.5	18.4	20.6
35–44	23.6	22.5	24.7	15.5	14.5	16.5
45–54	28.3	26.9	29.8	10.9	10.0	11.9
55–64	26.9	25.6	28.2	8.2	7.5	9.0
65 or older	28.8	27.5	30.0	7.0	6.3	7.8
Schooling (years)						
0–8	19.1	18.5	19.7	11.2	10.6	11.8
09–11	24.4	23.5	25.4	18.6	17.8	19.5
12 or more	36.3	34.9	37.8	14.5	13.4	15.6
Income (minimum wage) ^b^						
<1	18.0	17.3	18.6	14.1	13.5	14.8
≥1 a <3	26.2	25.3	27.0	15.0	14.3	15.8
≥3 a <5	35.8	33.7	38.1	14.4	12.7	16.2
≥5	41.7	39.2	44.2	10.4	9.0	12.1
Weight status						
Eutrophic ^c^	24.2	23.3	25.1	15.0	14.3	15.7
Pre-obesity ^d^	24.3	23.5	25.2	14.2	13.4	14.9
Obesity ^e^	22.3	21.2	23.4	13.1	12.3	14.0
Noncommunicable disease ^f^	25.6	24.8	26.5	9.9	9.3	10.4
Negative self-rated health	17.9	16.3	19.6	8.0	6.7	9.4
Total	23.7	23.2	24.2	14.3	13.8	14.8

CI 95%: 95% confidence interval; NHS: National Health Survey. ^a^ Weighted percentage to adjust the sociodemographic distribution of the NHS sample to the distribution of the general population (see Methodology). ^b^ Minimum wage in 2019: R$998; Excluded data from non-responders. ^c^ BMI < 25 Kg/m^2^ and BMI ≥ 25 Kg/m^2^. ^d^ <30 Kg/m^2^. ^e^ BMI ≥ 30 Kg/m^2^. ^f^ Presence of one or more chronic diseases (high blood pressure, diabetes, dyslipidemia, heart disease, stroke, cancer, kidney failure chronic disease). n = 88,531.

**Table 4 ijerph-21-01123-t004:** Frequency ^a^ of food consumption indicators (and PRc and PRa), according to prolonged screen time during leisure time. NHS, 2019.

Variables	<3 h/day (%)	≥3 h/day (%)	PRc	95%CI		PRa	95%CI	
Consumption of healthy foods (≥5 food groups)								
Exposure to TV	24.30	21.50	0.88 *	0.84	0.93	0.88 *	0.84	0.92
Exposure to electronic devices	24.80	19.90	0.80 *	0.76	0.85	0.86 *	0.82	0.92
Total screens	25.30	20.90	0.83 *	0.79	0.87	0.86 *	0.82	0.90
Consumption of unhealthy foods (≥5 food groups)								
Exposure to TV	13.50	17.20	1.27 *	1.19	1.37	1.37 *	1.28	1.47
Exposure to electronic devices	12.20	21.70	1.78 *	1.67	1.90	1.22 *	1.13	1.31
Total screens	11.70	18.70	1.60 *	1.50	1.71	1.33 *	1.25	1.42

^a^ Weighted percentage to adjust the sociodemographic distribution of the NHS sample to the distribution of the adult population of each city projected for each of the years (see Methodology). PRc: crude prevalence ratio; PRa: adjusted prevalence ratio; TV: television; NHS: National Health Survey); 95% CI: confidence interval; electronic devices: computer, cell phone, or tablet. * *p*-value ≤ 0.05. PRa: adjusted for sociodemographic characteristics (sex, age, schooling, income, race/color) and health (weight status, negative self-rated health, chronic disease). n = 88,531.

## Data Availability

The data are available for public access and use on the official IBGE website (https://www.ibge.gov.br/estatisticas/sociais/saude.html; accessed on 22 August 2024).

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
