# Peer review of "Leisure Screen Time and Food Consumption among Brazilian Adults"

_ijerph, 2024, doi:10.3390/ijerph21091123_

Round 1

Reviewer 1 Report

Comments and Suggestions for Authors

An interesting topic and important area of research. 

Abstract

·      The background section before the objectives is missing.

Introduction

·      This section should be strengthened- in its current state it does not give sufficient background or examination of the knowledge base in this area of research to justify why this research study is needed.

-              Some statistics are necessary to demonstrate the burden of the underlying issues i.e. unhealthy food consumption, sedentary lifestyles/ insufficient physical activity, obesity and/ or NCD prevalence.

·      The case for the gap in the literature needs to be clearly articulated:

-              You mentioned in line 38-39 that ‘unhealthy food consumption has been associated with sedentary behaviors, such as prolonged exposure to television and computers.’ However, this is not informative enough and leaves the reader asking more questions.

-              More information is needed on the current literature base and what is currently known in this space and then you should very clearly express the gap in the current literature that your research will fill.

Materials and methods

·      Data organization- more detail is needed on how healthy food and unhealthy food were categorized.

·      Also, on line 94 you say ‘ a score that corresponds to the sum of “yes” answers’- was there any measure of quantity? Because someone might have had a shred of lettuce and say ‘yes; I ate vegetables yesterday. (If scores did not include consideration of quantity this may need to be included as a limitation). 

Results

·      The table names are too wordy. Remove ‘(% and 95% CI)’ and ‘(>18years)’.

·      Ensure layout of the tables is consistent throughout- correct table 3 text alignment. 

Discussion

·      Overall, the discussion requires some re-working:

-              Repetition of results in the first paragraph- This paragraph should state the key findings without repeating the results. 

-              Some background information included in the discussion belongs in the introduction.

-              ‘The country’ has been mentioned a few times throughout- explicitly state the name of the country please.

-              More is needed on the implications of your findings.

-              Ensure the flow of the discussion is logical and coherent.

Study limitations 

·      I think it is also important to note that leisure screen time especially that of TV was limited to watching (in this study) and did not include playing (e.g. via gaming consoles such as PlayStation, X-Box, etc). 

·      Another limitation is also that the results report on data that was collected in 2019- (5-years ago) important to note.

Overall

·      When referring to how many years of schooling participants had- it is not always written in a correct way throughout the manuscript making it confusing to understand at times. You mean to describe the participants level of educational attainment. For example, on page 7, lines 212- 214- you are comparing participants with lower educational attainment with those of intermediate level and higher educational attainment (ensure you have defined what constitutes the different levels in the methods section).  Please correct and ensure consistency throughout to avoid misrepresentation of the data. 

·      References- many of your references are related to children and/ or adolescents- although inferences can be made, more focus is required on your target population (adults). 

Comments on the Quality of English Language

·      Some grammatical issues throughout- please revise grammar.

Author Response

Point-by-point response (Reviewer 1)

The reviewer raised several points that have been clarified by us, point-by-point, in the document below. We have revised the paper to address all the issues raised by both reviewers. We are available for any further clarifications if required.

Best regards, Authors

Abstract: The background section before the objectives is missing.

Authors: We insert the background section in the abstract.

Introduction:  This section should be strengthened- in its current state it does not give sufficient background or examination of the knowledge base in this area of research to justify why this research study is needed. Some statistics are necessary to demonstrate the burden of the underlying issues i.e. unhealthy food consumption, sedentary lifestyles/ insufficient physical activity, obesity and/ or NCD prevalence.

Authors: We restructured the introduction to expand the justification for conducting the study as requested by all reviewers. We included epidemiological information to support and understand the scope of the problem.

Introduction:  The case for the gap in the literature needs to be clearly articulated:  You mentioned in line 38-39 that ‘unhealthy food consumption has been associated with sedentary behaviors, such as prolonged exposure to television and computers.’ However, this is not informative enough and leaves the reader asking more questions. More information is needed on the current literature base and what is currently known in this space and then you should very clearly express the gap in the current literature that your research will fill.

Authors:  We have restructured the introduction to expand the justification for conducting the study as requested by all reviewers.

Materials and methods: Data organization- more detail is needed on how healthy food and unhealthy food were categorized.

Authors: We rewrote the indicators and inserted the questions that were used as a basis for constructing the indicators to make the construction of each one clearer.

Materials and methods:  Also, on line 94 you say ‘ a score that corresponds to the sum of “yes” answers’- was there any measure of quantity? Because someone might have had a shred of lettuce and say ‘yes; I ate vegetables yesterday. (If scores did not include consideration of quantity this may need to be included as a limitation). 

Authors: We have changed the way indicators are presented. We have inserted the question form and answer options to improve understanding. The use of 7 groups for healthy people is due to the grouping done between foods with similar characteristics, this was also done for UPFs. The information available refers to “any consumption on the past 24h.

Results: The table names are too wordy. Remove ‘(% and 95% CI)’ and ‘(>18years)’.

Authors: We made the requested changes to the table titles.

Results:  Ensure layout of the tables is consistent throughout- correct table 3 text alignment. 

Authors: Thanks for the suggestion. We have updated the table formatting.

Discussion:    Overall, the discussion requires some re-working:  Repetition of results in the first paragraph- This paragraph should state the key findings without repeating the results. 

Authors: We made a general re-working on the discussion to make it clearer and more targeted.

Discussion:  Some background information included in the discussion belongs in the introduction.

Authors: We made a general update to the discussion to make it clearer and more targeted.

Discussion:  ‘The country’ has been mentioned a few times throughout- explicitly state the name of the country please.

Authors: We updated the section and inserted the name of the country, when the term was mentioned.

Discussion:     More is needed on the implications of your findings.

Authors: We made a general update to the discussion to make it clearer and more targeted. Emphasis was given to the implications.

Discussion:    Ensure the flow of the discussion is logical and coherent.

Authors: We made a general update to the discussion to make it clearer and more targeted.

Study limitations :   I think it is also important to note that leisure screen time especially that of TV was limited to watching (in this study) and did not include playing (e.g. via gaming consoles such as PlayStation, X-Box, etc). 

Authors: We have inserted the limitation into the text.

Study limitations:     Another limitation is also that the results report on data that was collected in 2019- (5-years ago) important to note.

 Authors: We have inserted the limitation into the text as requested.

Overall: When referring to how many years of schooling participants had- it is not always written in a correct way throughout the manuscript making it confusing to understand at times. You mean to describe the participants level of educational attainment. For example, on page 7, lines 212- 214- you are comparing participants with lower educational attainment with those of intermediate level and higher educational attainment (ensure you have defined what constitutes the different levels in the methods section).  Please correct and ensure consistency throughout to avoid misrepresentation of the data. 

Authors: We updated the presentation format of the results regarding education throughout the text to avoid any confusion.

References- many of your references are related to children and/ or adolescents- although inferences can be made, more focus is required on your target population (adults). 

Authors: We updated the references cited in the text to target the study population.

Comments on the Quality of English Language:  Some grammatical issues throughout- please revise grammar.

 Authors: We carried out a complete review of the text, including grammatical review, with a view to improving it.

Reviewer 2 Report

Comments and Suggestions for Authors

Comments are in the Word document attached.

Comments on the Quality of English Language

The quality of the English language is decent but could be enhanced for better clarity. Some sentences were confusing. A thorough proofreading by someone proficient in English would significantly improve it.

Author Response

Point-by-point response (Reviewer 2)

The reviewer raised several points that have been clarified by us, point-by-point, in the document below. We have revised the paper to address all the comments made by both reviewers. We are available for any further clarifications if required.

Best regards, Authors

 In this manuscript, the authors investigate the associations between screen time and the quality of food consumed among Brazilian adults. The study is promising, presenting interesting findings with a very large sample size. Additionally, it explores a relevant topic for society, focusing on a population often neglected in the scientific community. However, the manuscript seems incomplete and lacks clarity, preventing me from recommending it for publication in its current form.

Firstly, the authors are inconsistent in their language, which makes the manuscript confusing. For example, they refer to their predictor (exposure) as “screen time > 3 hours” in some instances and as “extended period” in others. The introduction is too short and lacks proper contextualization for the study, failing to explain the study’s objectives. The methods section is unclear and lacks information on data collection and analysis. The results section could be better presented, particularly the tables. Finally, the discussion is disconnected from the rest of the manuscript and includes discussions that are out of scope.

In addition to the lack of clarity and proper presentation in the manuscript, I am not convinced that the authors chose the correct analysis for their study. They need to justify why they are running six different Poisson regressions, as it seems they are testing the same hypothesis multiple times, which could inflate type I error rates. A chi-square/contingency table test might be more appropriate if the authors want to keep dichotomous variables. Alternatively, they could run a logistic regression with food consumption as the outcome (more than 5 / less than 5 food groups) and include all control variables. The authors could also use the variables as continuous, particularly screen time. It is also unclear why they chose a Poisson distribution/model, as their study does not estimate occurrences over time but compares proportions of healthy/unhealthy eating in each group of screen media use. The authors need to change their statistical analysis or provide a strong rationale for their method.

The authors also need to update their references to support their rationale and discussion. Many references have broken links, inconsistent formatting, are not in English, or are not from international journals. Additionally, there are at least six self-citations. The authors need more up-to-date international studies to base their rationale and conclusions.

In the following pages, there are some specific issues in the manuscript that need to be addressed.

Title : Why are some words capitalized and others not?

Authors: The title is organized according to the journal's submission rules. Connectors are in lower case letters.

Abstract : Line 9 – “To analyze the association between leisure screen time and food consumption” sounds like the authors are analyzing associations between screen time and the frequency of eating, which is not exactly what the authors are investigating here. Maybe “type of food consumed” would be more appropriate? This comment applies to the rest of the manuscript.

Authors: We made the modification to the objective to try to make it clearer, as suggested by the reviewer.

"Thus, this study aimed to investigate the relationship between prolonged exposure to screen time during leisure time, using different types of screens (TV, computer, cell phone and tablet) and food consumption based on the pattern of consumption of healthy and unhealthy foods."

Abstract : Line 12 – I think using abbreviations makes the manuscript more confusing for the reader. CCT is not a standard abbreviation in the field which makes readers having to memorize what that is. I recommend using “portables” instead, as in “portable devices”. This comment applies to other unnecessary abbreviations throughout the manuscript, such as UPF, NCDs, etc. Whenever possible, please avoid those.

Authors: We have removed unusual abbreviations throughout the text.

Abstract : Line 22 – The authors must make it clearer what kind of population they are referring to in “Prolonged exposure…”. Brazilian adults?

Authors: We have made changes to the text to make it clearer. The manuscript analyzed prolonged screen time among Brazilian adults. The characteristics of adults with prolonged screen time were included in the results.

Introduction : The ideas explored in the introduction are interesting, but they need to be further explored as the introduction seems incomplete. The introduction does not: provide a good sense of the state-of-the-art in the literature, explain how this study contributes to the current literature, fully develop their ideas, and does not explain their main variables. For example, the authors do not define unhealthy food consumption (line 38), but they discuss its association with sedentary behaviour.

Authors: We restructured the introduction to expand the justification for carrying out the study as requested by all reviewers. As for the term unhealthy foods, we inserted the references and meaning of the term considered for this study.

Introduction : Line 32 – The authors write “Unhealthy food consumption stands out among the risk factors for the main causes of disease and mortality worldwide (1).”. This is vague and confusing. Please name at least some of the diseases or conditions. In addition, please provide a reference to a scientific paper or report, not an infographic on a website. Finally, “risk factor for the main causes” does not seem to make sense because how can unhealthy food consumption be a risk factor for the cause of a disease? It is either a risk factor or a main cause.

Authors: We made the requested changes to the text.

“Unhealthy food consumption, based on high consumption of ultra-processed foods increases the risk of noncommunicable diseases (NCD), such as cancer and cardiometabolic multimorbidity.”

Introduction : Line 39 – Confusing choice of words: “This association can be given by the increase in calorie consumption by promoting the passive (unaware) consumption of foods while sitting in front of a screen”. Perhaps “can be explained” is better? Line 42 – It is unclear how passive consumption increases the consumption of ultra-processed foods. It is also unclear how can someone be unaware that they are eating.

Authors: We re-wrote the sentence to make it clearer.

Introduction : Lines 44 to 47 – This whole sentence seems confusing to the reader. I am not sure what the authors meant. For example, I don’t understand what isolated indicators of unhealthy food consumption have to do with screen time. In another example “or even to food...” does not seem to make sense.

Authors: We apologize for the lack of clarity in the text. We restructured the introduction to expand the justification for carrying out the study as requested by all reviewers.

Introduction : Line 48 – “In an isolated sense, …” what does that mean? In the same sentence, the authors mention evidence shows an expressive increase in screen time in Brazil, but do not mention compared to when or what. Introduction : Line 50 – “this study sought to further investigate this relationship” it is unclear to what relationship are the authors referring to.

Authors: We apologize for the lack of clarity in the text. We restructured the introduction to expand the justification for carrying out the study as requested by all reviewers.

Method: Line 56 – The authors should specify the timeframe and location of data collection. Additionally, they need to provide a detailed description of the data collection methods and address the ethical considerations involved. This includes clarifying whether participants gave informed consent and if the study adhered to the Declaration of Helsinki guidelines.

Authors: We inserted the requested information into the methods.

Method: Line 77 – Please create a subsection in the method for the main independent variable (or predictor), another one for the dependent variable (outcome), and a third for control independent variables.

Authors: We inserted subsections into the methods as requested.

Method: Line 86 – Does “corresponds to the union” mean the sum of prolonged TV and portable consumption? If so, please make it clearer.

Authors: We apologize for the lack of clarity in the text. We rewrote the sentences referring to the construction of the indicators to make it clearer.

Method: Lines 88 to 104 – The authors need to make it clearer how the score of healthy and unhealthy food consumption was computed. For example, explain why the score varied from zero to seven for healthy and zero to ten for unhealthy. In addition, what exactly the answer “yes” stands for? Is it “yes” for each food group? Or “yes” for each period of time, participants ate the specific type of food.  Furthermore, how are researchers accounting for when participants ate both healthy and unhealthy food? Or when participants overeat?

Authors: We changed the presentation of the scores in the manuscript. We have inserted the question form and answer options to improve understanding. The use of 7 groups for healthy people is due to the grouping done between foods with similar characteristics, this was also done for UPFs. Unfortunately, information on quantities consumed was not collected, so we have included this information in the limitations. We do not consider consumption together. We know that the consumption of healthy foods and ultra-processed foods overlap, however, we seek to understand the association between the greater consumption of the groups independently. Furthermore, this is a score validated in previous studies.

References:
Costa C dos S, Sattamini IF, Steele EM, Louzada ML da C, Claro RM, Monteiro CA. Consumption of ultra-processed foods and its association with sociodemographic factors in the adult population of the 27 Brazilian state capitals (2019). Rev Saude Publica [Internet]. 2021 Jul 27;55:47. Available from: <https://www.revistas.usp.br/rsp/article/view/189149>.

Santin F, Gabe KT, Levy RB, Jaime PC. Food consumption markers and associated factors in Brazil: distribution and evolution, Brazilian National Health Survey, 2013 and 2019. Cad Saude Publica [Internet]. 2022;38(suppl 1). Available from: <http://www.scielo.br/scielo.php?script=sci_arttext&pid=S0102-311X2022001305005&tlng=en>.

Method: Line 90 – Please clearly define what unhealthy food consumption is.

Authors: We made changes to the section to make it clearer (including the definition of unhealthy food consumption).

Method: Lines 123 to 128 – The data analysis section does not provide information on how the chosen strategy helps to answer the research question or the reason the model was chosen. It is also not clear what are the variables in each model and how they were used. In addition, in the first sentence, the authors give the impression that screen time is also an outcome, when in fact it is the predictor.

Authors:  The requested information was added. Briefly, the approach was chosen because it allows for the investigation of the influence of each screen type, enabling the identification of any differences between them. The subsection was re-worked to make it clearer.

Results: Table 1 – The authors need to mention the Table in the text with a brief explanation of what it is. They could also make it clearer why they are using inferential statistics (95% CI) in a table describing their sample characteristics.

Authors: The data used came from the 2019 NHS. The NHS relies on a national representative sample of the Brazilian population. The description of the population made in table 1 characterizes the individuals interviewed (sample) based on the 2019 NHS. We reinforced this information in the methods.

Results: Line 149 – “Watching TV for extended periods was more frequent among the older age groups (65 years or older, 31.6%) and those with a lower schooling (23.9%)”. I understand that the 95% CI of some groups do not overlap, but if the authors did not conduct any inferential statistics to compare groups (chi-square or ANOVAs) I am not sure they can make such statements. If they did, they should make it clearer. The same applies to statements in the next paragraph (e.g., line 169).

Authors: Since no specific hypothesis was in test, we opted for an approach that could provide a wide range of comparisons between the groups. In this sense using the 95% CI provides a conservative approach (identifying differences through the non-overlapping of intervals). The information was added to the methods.

Results: Tables 2 and 3 – the layout of the tables could be improved for readability, as it can be challenging to read some numbers, particularly in table 2. Moreover, if the goal of the tables was to compare groups, I think the authors could present the effect size and p-value of inferential statistics testing for differences between groups.

Authors: Similarly to the case presented for Table 1, the idea to use the 95%CI was to provide a wide range of comparisons between the groups (since no specific hypothesis was in test). We opted to maintain the original format in the present version. The information was added to the methods section.

Results:Table 4 – If the alpha significance level was 5%, authors should flag p < .05 and not p <.001.

Authors: We apologize for the mistake. We have made the necessary changes.

Discussion: The discussion seems disconnected from the introduction and disproportionally long compared to other sections of the manuscript. It does not provide answers to the research questions and  does not discuss the findings using the current literature satisfactorily. Instead, in this section authors discuss topics that are out of the scope of their study.

Authors: We apologize for the lack of objectivity in the text. We re –worked the discussion to meet the reviewers' suggestions.

Discussion: Line 206 to 222 – The authors should refrain from drawing conclusions of group differences that were not tested and were part of the descriptive statistics of the manuscript. This part of the discussion seemed disconnected from the introduction. Authors should discuss how their findings answer their research question in this section, which they do not.

Authors: We apologize for the lack of objectivity in the text.  re –worked the discussion to meet the reviewers' suggestions.

Discussion: Line 208 – Please replace “population” with “sample” in this sentence and any other part of the manuscript meaning the sample of the study.

Authors: We changed to the text accordingly.

Discussion: Lines 232 to 236 – It is not clear how the current study explores those topics. These seemed out of scope.

Authors: We apologize for the lack of objectivity in the text.  re –worked the discussion to meet the reviewers' suggestions.

Discussion: Line 241 – Before the discussion, the authors never mentioned the “Brazilian Dietary Guidelines”. If this was part of the study design, it should be discussed in the introduction or method.

Authors: We appreciate the suggestion. We included the context of the Brazilian Dietary Guidelines in the methods, although it was not directly analyzed, the classification of the foods used in the study is based on the recommendations of the Guide, considering the consumption of natural and minimally processed foods as a healthy diet and the consumption of ultra-processed foods as an unhealthy diet.

Discussion: Lines 223 to 243 – This paragraph seemed more relevant to the introduction and not the discussion.

Authors: We moved the most important part of the paragraph to the introduction and removed the content that was not highlighted.

Discussion: Lines 244 to 257 – This paragraph seemed out of the scope of this manuscript and suggests that the authors’ own study is not useful in the sense it cannot be compared to other studies.

Authors:  We have reworded the discussion to address reviewers' suggestions.

Discussion: Lines 259 to 268 – In the previous paragraph the authors made the case that their results could not be compared to other studies, just to contradict themselves by comparing their results in this paragraph. Moreover, they compare their findings to studies with a decade or more of difference. This comparison can be biased, as screen media technology has quickly evolved in the past decade, individuals replaced TV time with smartphones and computers over the years.

Authors:  We have reworded the discussion to address reviewers' suggestions.

Discussion: Line 286 – The authors mention “It is clear that the screen exposure time is only part of the problem”. Yet, to assert that something is clear, authors must present evidence, which they do not.

Authors:  We have reworded the discussion to meet the reviewers' suggestions.

Discussion: Lines 293 to 313 – It is unclear why the authors dedicated a massive paragraph to discuss interventions given that is out of the scope of their study.

Authors: We updated the section, reducing its content and expanding the scope of the discussion of interventions beyond screen time to also include interventions related to food.

Discussion: Lines 324 to 328 – Those are not limitations. In fact, the study being conducted in Brazil is a Strength of this study.

Authors: We reinforce the study being carried out in Brazil as one of the strengths.

Discussion: Lines 332 to 333 – This one-sentence paragraph is unclear and seems out of place.

Authors: We redid an important part of the discussion to meet the reviewers' suggestions. We hope that the study has become more objective.

Discussion: Last paragraph – This seems to be the conclusions, but it is under the study limitation section.

Authors: We inserted the conclusion section into the text to separate this specific section.

Round 2

Reviewer 1 Report

Comments and Suggestions for Authors

Overall, the authors have addressed the comments well and the quality of the manuscript has been significantly improved. 

Suggestions:

1. In the introduction- line 49- please fix the reference (Swineburn, 2019) so that the format is consistent with the referencing style used throughout the manuscript. 

2. The discussion- overall good- but I think you have room for a bit more critical discussion on the implications of your findings- linking to other research. This will strengthen the quality of your paper.

Author Response

Point-by-point response (Reviewer 1)

We have addressed each of the reviewers' comments in detail, as outlined in the document below. The manuscript has been revised to address all concerns raised by both reviewers. Additionally, a thorough review of the text was conducted to improve the quality of the writing. We remain available for any further clarification, should it be necessary.

Sincerely,
Authors

Overall, the authors have addressed the comments well and the quality of the manuscript has been significantly improved. 

Suggestions:

  1. In the introduction- line 49- please fix the reference (Swineburn, 2019) so that the format is consistent with the referencing style used throughout the manuscript. 

Authors: We have updated it to the appropriate format. Thank you for pointing it out.

  1. The discussion- overall good- but I think you have room for a bit more critical discussion on the implications of your findings- linking to other research. This will strengthen the quality of your paper.

Authors: Thank you for highlighting these points. We have revised the manuscript to better address the critical aspects and their implications.

“The results of our study have important public health implications. Identifying specific risk groups, such as women, the elderly and individuals with lower levels of education and income, who are more likely to use screens for a long time and, consequently, may develop inadequate eating habits, is essential for monitoring risk groups for NCDs and their risk factors in the country (34). These potential risk groups have been observed in other studies, where sociodemographic factors like age and gender were found to moderate the association between food consumption and screen time (7). These findings suggest the need for targeted interventions that consider the sociodemographic characteristics of these groups to maximize the effectiveness of health promotion strategies (7). However, further analysis is needed to clarify this as it was not the central focus of this study. It should be noted that such interventions could be effective in preventing NCDs, such as obesity, diabetes, and cardiovascular diseases, given the relationship observed in previous studies (26-29).

The negative impact of prolonged screen time on diet quality highlights the importance of public policies that encourage reducing screen time as a strategy to improve dietary patterns and, by extension, the health of the population. While prolonged screen time is only one aspect of the issue related to unhealthy food consumption in the context of the current food system (3), it remains a significant risk factor for several diseases (26-29). In our study, we observed a direct association between the consumption of unhealthy foods and all types of screen time, as well as an inverse association with the consumption of healthy foods. Therefore, it's crucial to consider interventions that effectively reduce screen time and address current dietary patterns, as these measures are essential for tackling the broader public health challenges associated with both screen time and unhealthy eating habits. In a systematic review conducted in 2011 (7), this relationship was observed among specific food groups, such as the inverse relationship between TV time and fruits and vegetables consumption, and a direct relationship with the snacks and sweetened beverages consumption among children, adolescents, and adults (7). However, screen time on electronic devices, has not been well reported (7), particularly between adults, largely due to the recent increase in the use of this type of screen by the population (15-17), with the association between the consumption of snacks, fried and sweet foods and high screen time on electronic devices in children and adolescents (14) being more widely reported.

In this sense, emphasizing interventions to reduce screen time and sedentary behavior is crucial to addressing the broader public health challenges associated with screen time and unhealthy eating habits. Interventions aimed at reducing screen time and sedentary behavior often focus on both decreasing overall screen time (8) and promoting leisure physical activity (7, 8). The Brazilian Physical Activity Guidelines recommend moving for at least 5 minutes every hour to enhance quality of life (35). Improve diets, strategies such as taxing ultra-processed foods, regulating advertising on TV and other devices, and encouraging healthy eating through subsidies can be effective (3, 36). Additionally, reducing the link between prolonged screen time and food consumption is reinforced by various studies (6, 7, 11-14) and Brazil's dietary guidelines, which suggest avoiding eating in front of screens and making mealtimes a shared family experience at the table (22).”

Reviewer 2 Report

Comments and Suggestions for Authors

The presentation of the manuscript has greatly improved. Particularly in the discussion. Yet, the introduction seems still incomplete, not satisfactorily discussing the context of the international literature or explaining how this study adds new information to the literature. More importantly, the authors did not address important issues I raised in my previous review. For example, the authors did not improve their statistical analysis, nor did they justify their methodology of using Poisson regression models. Please see my previous review for detailed explanation. While I feel the authors did a good job improving the presentation of the manuscript, I feel that they failed to address important points raised previously.

Comments on the Quality of English Language

The language use has improved since the last review, but the manuscript could benefit of a good read through.

Author Response

Point-by-point response (Reviewer 2)

We have addressed each of the reviewers' comments in detail, as outlined in the document below. The manuscript has been revised to address all concerns raised by both reviewers. Additionally, a thorough review of the text was conducted to improve the quality of the writing. We remain available for any further clarification, should it be necessary.

Sincerely,
Authors

The presentation of the manuscript has greatly improved. Particularly in the discussion. Yet, the introduction seems still incomplete, not satisfactorily discussing the context of the international literature or explaining how this study adds new information to the literature.

Authors: We have included additional information in the introduction to provide a clearer explanation of how this study contributes new information to the existing body of research.

“The relationship between screen time and food consumption has been explored by several researchers worldwide (6, 7, 11-14). The main sedentary behavior analyzed was prolonged TV watching (6, 7, 11-13), particularly due to the culture over the past decades of spending time in front of this type of screen while eating (7). However, over the years, screen time on other electronic devices, such as computers, cell phones, tablets, and video games, has increased considerably, especially among younger individuals in the population (15-17). In this context, although some studies are already analyzing the transition from prolonged TV viewing to other electronic devices, particularly among children and adolescents (8, 14), our understanding of the relationship between different types of screen time and food consumption among adults is still incomplete.

Furthermore, previous studies have often examined the association between screen time and specific food consumption indicators, such as fruits, vegetables, snacks, or sweetened beverages, in isolation (6, 7, 11, 12). This narrow focus overlooks broader dietary patterns that emerge when multiple food groups are considered together. To address this gap, our study employs a simplified dietary scoring model, which has been utilized in various contexts to assess and monitor dietary patterns (18). This approach not only enhances our understanding of how screen time influences overall diet quality but also fills a crucial gap in the literature by analyzing the association between screen time and comprehensive dietary patterns in adults.”

More importantly, the authors did not address important issues I raised in my previous review. For example, the authors did not improve their statistical analysis, nor did they justify their methodology of using Poisson regression models. Please see my previous review for detailed explanation. While I feel the authors did a good job improving the presentation of the manuscript, I feel that they failed to address important points raised previously.

Authors: While we acknowledge the reviewer's critique as valid, we would like to emphasize that explanations/justifications were provided for all the issues raised, as detailed in the previous letter. We regret that those were not sufficient. Nevertheless, a comprehensive rationale for the use of Poisson regressions (instead of logistic regressions) has been added to the methods section, along with a more detailed explanation of the dependent, independent, and confounding variables in the models.

“Poisson regression models (with robust variance) were applied to analyze the association between the indicators of healthy and unhealthy food consumption (dichotomous dependent variable) and indicators of in prolonged screen time during leisure time (dichotomous independent variables). Both crude (PRc) and adjusted prevalence ratio (PRa) were calculated, with adjustments made for sociodemographic (sex, age, schooling, income, race/color) and health (weight status, negative self- rated health, chronic disease) variables in the models.

 Poisson models are a useful alternative to logistic regression for dichotomous outcomes in cross-sectional studies since the latter may overestimate the true association, especially when the prevalence of the dependent variable is high (over 10-20% (25), as observed in the results of the present study). The confounding variables (mentioned above) were selected based on their relationships with both the dependent and independent variables used in the regression models (data not shown). In the adjusted models, all confounding variables were simultaneously included, allowing for the observation of the effect of the independent variable (prolonged screen time during leisure and its variations) on the dependent variables (related to food consumption). Three different models were employed for each outcome, independently investigating the target association for prolonged screen time in TV, in electronic devices and combining the use of both types of screens. This approach was chosen because it allows for the investigation of the influence of each screen type, enabling the identification of any differences between them.”

Reference: Barros, A. J. D., & Hirakata, V. N. (2003). Alternatives for logistic regression in cross-sectional studies: an empirical comparison of models that directly estimate the prevalence ratio. BMC Medical Research Methodology, 3, 21. DOI: 10.1186/1471-2288-3-21.

The language use has improved since the last review, but the manuscript could benefit of a good read through.

Authors: We appreciate the feedback. We have conducted a thorough review to enhance the language of the manuscript.